# Acceptance of Human Papillomavirus (HPV) vaccine among the parents of eligible daughters (9–15 years) in Bangladesh: A nationwide study using Health Belief Model

Mohammad Delwer Hossain Hawlader[1], Fahima Nasrin Eva[1,2]*, Md. Abdullah Saeed Khan[2,3], Tariful Islam[1,2], Umme Kulsum Monisha[2,4], Irin Chowdhury[1,2], Rifat Ara[2,5], Nur-E-Safa Meem[1,2], Mohammad Ali Hossain[2,6], Arpita Goutam[1,2], Tahmina Zerin[1,2], Nishat Alam[1,2], Rima Nath[1,2], Shamma Sifat[1,2], Sayla Sultana[1,2], Mosammat Sadeka Sultana[1,2], Sumit Kumar Saha[1,2], Naifa Enam Sarker[1,2], Mohammad Hayatun Nabi[1], Mohammad Lutfor Rahman[7]

1 Department of Public Health, North South University, Dhaka, Bangladesh, 2 Public Health Promotion and Development Society (PPDS), Dhaka, Bangladesh, 3 National Institute of Preventive and Social Medicine (NIPSOM), Dhaka, Bangladesh, 4 Mandy Dental College & Hospital, Dhaka, Bangladesh, 5 Institute for Global Health, University College London, London, United Kingdom, 6 Ibn Sina Medical College Hospital, Kallyanpur, Dhaka, Bangladesh, 7 Institute of Statistical Research and Training (ISRT), University of Dhaka, Dhaka, Bangladesh

* fahima.eva@northsouth.edu

## Abstract

### Background

To align with the 2030 vision of the World Health Organization (WHO) to ensure 90% of girls receive the HPV vaccine before turning 15, Bangladesh has recently started the (HPV) vaccine campaign nationwide. Therefore, our study aimed to assess the level of its acceptance among parents of eligible daughters in Bangladesh.

### Methods

This nationwide cross-sectional study was conducted among the self-identified parents of daughters in the age group of 9–15 years between June 28 and August 2, 2023. A total of 2,151 parents were conveniently selected from all eight divisions of Bangladesh. Data was collected through face-to-face interviews using a semi-structured questionnaire. The Health Belief Model was used to appraise respondents' beliefs concerning HPV and its vaccination. Adjusted odds ratios (AOR) with a corresponding 95% confidence interval (CI), and a p-value of <0.05 was considered statistically significant. R Studio (Version 2023.09.0+463) was used as an interface for data analysis, while R (the programming language) was used for statistical computations.

### Results

The mean age of the study participants was 38.18 (±5.86) years. The overall acceptance rate of the HPV vaccine was 86.61% (95%CI: 85.09–88.02). The vaccine acceptance level

**Data Availability Statement:** All relevant data are within the manuscript and its Supporting Information files.

**Funding:** The author(s) received no specific funding for this work.

**Competing interests:** The authors have declared that no competing interests exist.

was positively associated with all the domains of the HBM (p<0.001), except in the perceived barrier domain (p = 0.489). After adjustment for other factors, it was found that higher age was associated with a decreased acceptance (AOR: 0.92; 95% CI: 0.89–0.95). The urban residents exhibited 42% lower odds of vaccine acceptance than rural (AOR: 0.58; 95% CI: 0.36–0.92). Similarly, participants of the middle-income group had 44% lower odds than the lower-income group (AOR: 0.56; 95% CI: 0.32–0.97).

## Conclusion

Our study found a reasonably good level of acceptance of the HPV vaccine among the parents of eligible daughters. Multiple factors such as younger age, urban residence, belonging to the middle income group, history of regular routine health check-ups, knowledge of cervical cancer, positive perception about benefits of the vaccine, and positive cues to actions were associated with HPV vaccine acceptance.

## Introduction

The global surge in HPV (Human Papillomavirus) infections is a critical concern today. HPV contributes to around 500,000 annual cancer cases, including cervical, vulvar, anal, penile, and oropharyngeal cancer [1]. Ninety-nine percent of cervical cancer cases in females result from sexually transmitted HPV [2]. Cervical cancer is the fourth most common cancer among women, with a global age-standardized incidence rate of 13.3/100,000 women and a mortality rate of 7.3/100,000 women [3]. In Bangladesh, cervical carcinoma is the second leading malignancy in women, causing 12,000 new cases and over 6,000 deaths annually [4]. HPV, specifically high-risk strains HPV-16 and 18, accounts for over 70% of cervical cancer cases worldwide [5].

The vaccination against HPV emerges as a pivotal strategy for significantly reducing cervical cancer incidence, offering both bivalent and quadrivalent options. While bivalent vaccines primarily protect against high-risk HPV varieties (HPV 16, 18), quadrivalent vaccines target both low and high-risk HPV strains (HPV 6, 11, 16, and 18) [6]. Moreover, screening methods like the Pap smear, visual examination with acetic acid, and self-sampling for HPV DNA testing are effective in cervical cancer prevention [7]. However, eliminating sexual risk factors and vaccination are the primary prevention methods [8]. The WHO recommends vaccinating girls aged 9–14 as the primary target group, with females ≥15 as the secondary group [9].

Though vaccination is an effective way to reduce cervical cancer, there exists a distinction in vaccination coverage based on the development status of the country [10]. In high-income countries, the coverage is almost 80% [11]. Although data on low-and-middle-income countries and low-income countries are scarce, one study reported that in low-and-middle-income countries, the vaccination coverage is about 64% [7].

With a vision for 2030, WHO aims to ensure that 90% of girls receive the HPV vaccine before turning 15 [12]. The success of this ambitious goal depends largely on the broad acceptance of the vaccine, a factor closely tied to knowledge and awareness about cancer prevention measures among both the target population and their parents [6, 13]. For the HPV vaccine, the acceptance of parents is essential as the recipient groups are minors dependent on their parents for decision-making. However, vaccine acceptance among parents is not universal. They often refuse to vaccinate their children on philosophical or religious grounds or out of

concern for adverse outcomes [14]. For instance, two studies conducted in Ethiopia found that 81.3% and 94.3% of the parents and guardians of eligible daughters were willing to vaccinate their children, respectively [5, 15]. In China, a meta-analysis reported a pooled acceptance of 55.29% among parents of primary and junior high school students only [16]. The study also noted that parental awareness is a significant determinant of HPV vaccine acceptance. The Government of Bangladesh, with support from UNICEF, the Vaccine Alliance (Gavi), and WHO, initiated a groundbreaking Human Papillomavirus (HPV) vaccination campaign on October 2, 2023 [17]. However, there is a lack of comprehensive, nationwide cross-sectional studies regarding the acceptance of the HPV vaccine among the parents of adolescent girls in Bangladesh. Therefore, our nationwide study aimed to assess the level of acceptance of HPV vaccines among parents of adolescent girls in Bangladesh before the national rollout, with the ultimate goal of ensuring the program's success.

## Methodology

### Study design and participants

We conducted a nationwide cross-sectional study among the self-identified parents of daughters in the age group of 9–15 years in Bangladesh. Parents with foreign nationality and/or those who were diagnosed and were taking medication for mental health illness were excluded. To ensure representative sampling, we determined division-specific sample sizes using the 2022 Population & Housing Census [18], resulting in a total adjusted sample size of 2,160 participants, while considering an 80% vaccine acceptance among parents based on the existing literature [9, 19–21] and 10% non-response. We collected samples conveniently from all eight divisions (Dhaka, Mymensingh, Chattogram, Sylhet, Rajshahi, Khulna, Rangpur, and Barisal) of Bangladesh and covered 42 out of 64 districts between June 28 and August 2, 2023. The division-wise distribution of samples is listed in the supplementary table (**S1 Table**). The overall response rate, considering those who actively participated in the survey, is approximately 90%. Of the 2160 eligible participants who agreed to participate, 2151 completed the entire questionnaire (completion rate: 99.58%); incomplete questionnaires were excluded from the analysis.

### Study procedure

A team of 20 competent and trained public health graduates and students collected data from the respondents through face-to-face interviews. They were instructed to speak with as many parents as they could, regardless of their backgrounds, since a convenience sampling technique was used. To avoid linguistic hurdles, the team allocated and trained interviewers according to their locality. Participants were approached in public settings such as hospitals, schools, pharmacies, food markets, roadways, offices, and houses. Along with interacting with the general public, relatives, friends, neighbours, and colleagues of the respondents were invited for the interview. To make the questionnaire easier for the study participants to understand, we translated the difficult terms into a straightforward, native version. While answering the questionnaire, our interviewers provided explanations to participants to assist them understand certain items. Participation was entirely voluntary, and written informed consent was obtained prior to enrolment in the study.

### Questionnaire

We used a semi-structured questionnaire to conduct face-to-face interviews. The questionnaire consisted of sections on 1) socio-demographic information, vaccination history of

daughters, and health behaviour (21 items); 2) knowledge and information sources related to HPV and the HPV vaccine (11 items); 3) HBM constructs surrounding HPV and HPV vaccine (24 items); 4) acceptance of HPV vaccine (2 items). After a comprehensive literature review, an English draft questionnaire covering the HBM model was developed based on previous studies [22–24]. The questions were developed to cover the domains of the HBM model, which was proposed by a group of social psychologists at the US Public Health Service [25]. We modified and adjusted the framing and wording of questions to reflect Bangladesh's socio-cultural and healthcare context. Following that, a Bangla translation of the questionnaire was made. Translated versions were validated by repeated revisions and agreement among the researchers. Forward-background translation by certified translators and review by relevant experts were outside the scope of the investigation due to the time-sensitive nature of the study. However, in two steps, the questionnaire's face validity was ensured. First, we shared the tool with the study supervisor and other researchers in order to obtain expert feedback on its clarity, relevance, and significance. Second, before commencing the actual data collection, a pre-test involving 28 parents of daughters aged between 9–15 years old was undertaken. To enhance participant comprehension and bolster both face- and construct validity, certain questions were revised or clarified in the questionnaire. For the pilot study, we chose participants from a wide range of socioeconomic backgrounds. In order to maintain consistency with recent literature, participant amendments were taken into account and incorporated into the survey. Following a thorough discussion, the authors finalized the questionnaire (58 items) and distributed it for the purpose of the study. Per participant, the questionnaire took about 10 to 12 minutes to complete. The complete English version of the questionnaire is available in the supplementary materials (S1 File).

## Variables and measurement

**Socio-demographic information, previous vaccination history of daughters, religious belief, and health profile.** The survey categorized participants as either mothers (i.e., females) or fathers (i.e., males). Comprehensive personal information encompassing address, age, marital status, religion, residence, education, occupation, average monthly family income, family size, number of children, and daughters' ages were collected. Binary questions assessed the healthcare worker status and daughters' educational enrollment. Childhood vaccinations were classified as received or not. Responses indicating "yes all" or "yes some" for other recommended vaccines were grouped as "yes," whereas "no" or "not sure" responses were categorized as "others." Academic institution type (government, private, madrasa) and location (rural, semi-urban, urban) were assessed as nominal variables. Health check-up frequency was measured on an ordinal scale: "never," "<1 year," "1–2 years," "2–5 years," and ">5 years".

**Knowledge and source of information about HPV and HPV vaccine.** Knowledge was defined as the state of being aware of the term of interest in this study. Respondents' knowledge and source of information about HPV, HPV vaccine, cervical cancer, and cervical cancer-related vaccines were assessed through eight questions. First, they were asked if they had heard about HPV to investigate their awareness of HPV. If respondents answered affirmatively in the awareness-related question, their sources of information were sought. Knowledge about the HPV vaccine, cervical cancer, and cervical cancer-related vaccines was sought in a similar fashion.

**HBM constructs surrounding HPV and HPV vaccine.** The Health Belief Model was employed to appraise respondents' beliefs concerning HPV and its vaccination. This evaluation encompassed five aspects: perceived susceptibility (5 items), severity (5 items), benefits (5 items), barriers (5 items), and cues to action (4 items) related to HPV and its vaccine.

Respondents' perceptions of susceptibility, severity, benefits, and barriers were rated on a five-point Likert scale, ranging from "Strongly Disagree" (1 point) to "Strongly Agree" (5 points). Meanwhile, cues to action were measured using a binary 'Yes'/'No' scale, with 'Yes' scored as 1 and 'No' as 0.

**HPV vaccine acceptance.** HPV vaccine acceptance was evaluated using two questions on vaccinating daughters and response to a government-provided free vaccination, employing a 3-point scale ('Yes', 'No and 'Not Sure').

## Statistical analysis

All data were checked for completeness, outliers, and assumption violations prior to analysis. Descriptive statistics were used to characterize the socio-demographic information of the study participants. The responses to the questions about acceptance of the HPV vaccine- 'No' and 'Not sure' were merged to produce one response- 'No' for statistical analysis. In HBM domains, answers to each individual question were first added for perceived vulnerability, perceived severity, perceived benefit, and perceived barrier domains. This gave a total score between 5 to 25. Then, a score of $\geq 20$ was considered a 'positive', and $<20$ was considered a 'negative' response for each domain. For the cues to action domain, a positive answer in one of the four questions was considered positive for the overall domain. Knowledge and beliefs about HPV and its vaccines and their association with the acceptance of the vaccine were analyzed using the Chi-square test, Fisher's exact test, and Welch's Two Sample t-test where appropriate. Multivariable binary logistic regression analysis was performed to identify the determinants of HPV vaccine acceptance among parents' sociodemographic factors, health check-up frequency, and knowledge and belief regarding the HPV and HPV vaccine and cervical cancer. Vaccine acceptance was dichotomized by grouping "no" and "not sure" into one category. Thereby, the emphasis was on the acceptance of the vaccine rather than uncertainty or denial. The model performance was measured using Nagelkerke's pseudo-R-squared (0.368), Receiver Operating Characteristics Curve (Area under the curve: 0.8471), and Hosmer-Lemeshow goodness of fit test ($\chi^2$ = 2145, df = 8, p $<$0.001). Although the model was not a good fit, we kept the model because of its overall significance over a null model (p$<$0.001) and because we were interested in identifying significant determinants of vaccine acceptance rather than the predictive accuracy of the overall model. Variables that were significant in bivariate analyses were included in the multivariable logistic model. Adjusted odds ratios (AOR) were expressed with a corresponding 95% confidence interval (CI). A p-value of $<$0.05 was considered significant for statistical tests. R Studio (Version 2023.09.0+463) was used as an interface for data analysis, while R (the programming language) was used for statistical computations.

## Ethical statement

All the procedures were carried out in accordance with the ethical guidelines of North South University's Institutional Review Board (IRB)/Ethical Review Committee (ERC) (Approval no: 2023/OR-NSU/IRB/0507). Wherever applicable, the ethical standards outlined in the 1964 Helsinki Declaration and its subsequent amendments, or comparable ethical standards, were followed [26]. During the face-to-face interviews, we obtained written informed consent from all the participants involved in the study. Before providing their written consent, all the participants were informed that their participation was entirely voluntary and that they could withdraw participation at any time. Additionally, they were also informed that all data would be presented on a group level and that only the researchers would have access to it.

## Results

Table 1 represents the characteristics of the respondents. The mean age of the study participants was 38.18 (±5.86) years. Of all, 81.40% were female, and 93.86% lived with their spouse. On average, participants had 11.37 (±4.51) years of education. Most of the participants were house-wives (63.41%), while only 8.51% were healthcare workers, 51.60% of the respondents lived in an urban area, and 27.98% of households had a monthly income of 35,001–50,000 Bangladeshi Taka (BDT). Joint families (57.12%) were more common than nuclear families (42.88%). About 31.19% of respondents had done regular health check-ups. A significant portion of the population was aware of cervical cancer (80.15%). However, only 22.55% of respondents heard about Human Papillomavirus. Approximately 48.72% of participants knew about the cervical cancer vaccination, while only 22.32% knew about the HPV vaccine. About 21.57% of participants reported feeling vulnerable due to HPV infection, 19.39% felt the severity of the HPV infection and its health risk, 38.45% thought that the vaccination was beneficial, only 0.79% thought that there were barriers that would prohibit them from giving their daughters the HPV vaccine, and 41.79% of respondents had the urge to take action against cervical cancer. The overall accep-tance rate of the HPV vaccine among the participants was 86.61% (Fig 1).

Table 2 outlines participants' responses to health belief model-related questions. Regarding the Perceived Vulnerability domain, a significant portion of respondents agreed that HPV can cause sexually transmitted diseases (39.19%), can lead to condyloma/genital warts (66.20%), poses a risk for young women (37.94% agreed), represents a serious health concern (48.35%), and expressed concern about their child contracting HPV (47.28%). On the Perceived Severity domain, 24.17% of participants agreed that individuals with HPV might not exhibit symptoms, while the majority of the participants remained neutral (68.85%). Concerning the seriousness of conditions, a significant majority agreed (55.00%) that HPV-associated cervical cancer is a serious health concern. Furthermore, a notable portion of participants (49.37%) agreed regard-ing the possibility of HPV-associated cervical cancer occurring in middle age. Regarding the Perceived Benefits domain, the majority of the participants (64.71%) agreed that they trust vac-cinations, citing ongoing research as a reason for their trust. Regarding cervical cancer preven-tion, a considerable proportion agreed (54.21%) that the HPV vaccine would effectively prevent cervical cancer in their daughters. Regarding HPV transmission, 59.65% agreed vacci-nated girls are less likely to contract HPV than unvaccinated girls. Moreover, 50.26% of the participants agreed that HPV vaccination increases awareness of sexually transmitted diseases. On the Perceived Barrier domain, a substantial portion of participants strongly disagreed with several barriers. Specifically, many disagreed (67.46%) with the notion that vaccination should be avoided because of potential pain. In terms of concerns related to adverse effects, a signifi-cant number disagreed (54.81%) with this barrier, suggesting that they do not view adverse effects as a significant obstacle. Similarly, participants largely disagreed (60.25%) with the state-ment that the vaccine's requirement of two injections was a barrier. Regarding the perception that the HPV vaccine is new and requires waiting, participants varied in their responses, with a higher proportion agreeing (36.91%) and a notable proportion disagreeing (24.97%). Finally, regarding the cost barrier, a good number of participants (32.78%) agreed that they would vac-cinate their daughters despite the cost.

In response to cues to the action domain, individuals demonstrated diverse cancer-related backgrounds. While a minority disclosed their own experiences with cancer (1.53%) or cervi-cal cancer (1.49%), a notable majority acknowledged a cancer history (35.33%) or cervical can-cer history (17.29%) within their circle of friends or family (Fig 2).

The association between participants' characteristics and HPV vaccine acceptance for their daughters is illustrated in Table 3. The average age of the parents willing to vaccinate their

**Table 1. Characteristics of the respondents (n = 2,151).**

| Characteristic | n (%) |
|---|---|
| Age (years) | 38.18 ±5.86 |
| Sex | |
| Female | 1,751 (81.40) |
| Male | 400 (18.60) |
| Marital Status | |
| Living with spouse | 2,019 (93.86) |
| Living without spouse | 76 (3.53) |
| Others | 56 (2.60) |
| Religion | |
| Islam | 1,784 (82.94) |
| Hindu | 337 (15.67) |
| Buddhist | 4 (0.19) |
| Christian | 26 (1.21) |
| Residence | |
| Rural | 783 (36.40) |
| Semi-urban | 258 (11.99) |
| Urban | 1,110 (51.60) |
| Years of Education | 11.37 ±4.51 |
| Occupation | |
| Job | 511 (23.76) |
| Business | 186 (8.65) |
| Housewife | 1,364 (63.41) |
| Others | 90 (4.18) |
| Health Care Worker | 183 (8.51) |
| Monthly household income (BDT) | |
| < = 20000 | 547 (25.47) |
| 20001–35000 | 569 (26.49) |
| 35001–50000 | 601 (27.98) |
| >50000 | 431 (20.07) |
| Number of family members | 5.00 (4.00–6.00) |
| Family Type | |
| Nuclear | 922 (42.88) |
| Joint | 1,228 (57.12) |
| Routine Health Checkup | |
| Regular | 671 (31.19) |
| Irregular | 890 (41.38) |
| Never | 590 (27.43) |
| Heard about the Human Papillomavirus | 485 (22.55) |
| Heard about the HPV Vaccine | 480 (22.32) |
| Heard about Cervical Cancer | 1,724 (80.15) |
| Heard about Cervical Cancer Vaccination | 1,048 (48.72) |
| Perceived Vulnerability | |
| Negative | 1,687 (78.43) |
| Positive | 464 (21.57) |
| Perceived Severity | |
| Negative | 1,734 (80.61) |
| Positive | 417 (19.39) |

(*Continued*)

**Table 1.** (Continued)

| Characteristic | n (%) |
|---|---|
| Perceived Benefit | |
| Negative | 1,324 (61.55) |
| Positive | 827 (38.45) |
| Perceived Barrier | |
| Negative | 2,134 (99.21) |
| Positive | 17 (0.79) |
| Cues to Action | |
| Negative | 1,252 (58.21) |
| Positive | 899 (41.79) |

Continuous data was expressed as Mean ±SD and Median (IQR)

girls was significantly lower than those who did not accept the vaccine (p<0.001). Vaccine acceptance was significantly higher among mothers (i.e., female) (87.95%) than fathers (i.e., male) (p<0.001). The respondents living with their spouses were notably more interested in vaccinating their daughters than those living without their spouses. (p = 0.027). Semi-urban and rural dwellers were substantially more receptive to vaccinations than others. (p = 0.018). A notable positive association between education level and the acceptance of vaccines was found (p<0.001). Job holders and healthcare workers showed a higher inclination to get their daughters vaccinated compared to their other groups (p<0.001). The majority of the parents who expressed acceptance of the vaccine had relatively lower monthly family income (p = 0.003) and belonged to the nuclear family (p<0.001). Individuals with a history of regular (93.89%)

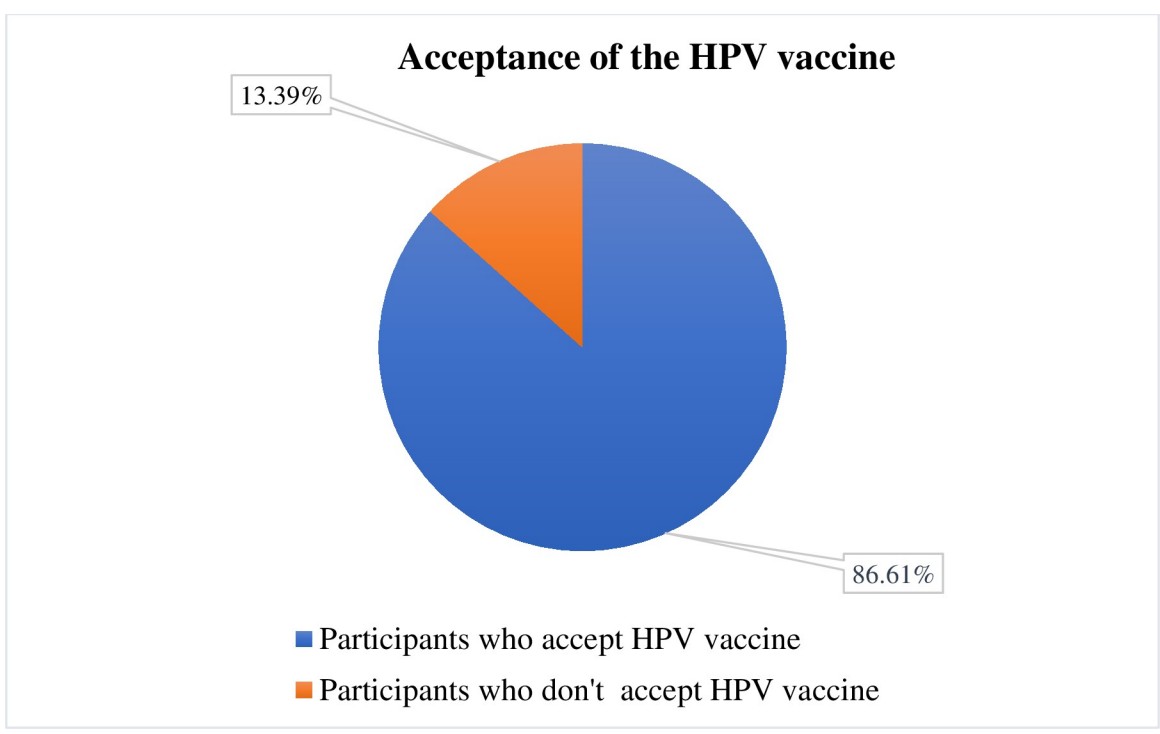

**Fig 1. Acceptance of the HPV vaccine.**

**Table 2. Distribution of responses to health belief model-related questions (n = 2,151).**

| Characteristic | | | | | n (%) |
|---|---|---|---|---|---|
| | Strongly Disagree | Disagree | Neutral | Agree | Strongly Agree |
| **Perceived Vulnerability** | | | | | |
| HPV can cause sexually transmitted diseases. | 1 (0.05) | 29 (1.35) | 1,175 (54.63) | 843 (39.19) | 103 (4.79) |
| HPV can cause condyloma/genital warts. | 1 (0.05) | 41 (1.91) | 1,424 (66.20) | 613 (28.50) | 72 (3.35) |
| There is a risk for young women to contract HPV. | 4 (0.19) | 44 (2.05) | 1,204 (55.97) | 816 (37.94) | 83 (3.86) |
| HPV infection is a serious health concern. | 4 (0.19) | 18 (0.84) | 870 (40.45) | 1,040 (48.35) | 219 (10.18) |
| I worry that my child might get HPV. | 7 (0.33) | 121 (5.63) | 860 (39.98) | 1,017 (47.28) | 146 (6.79) |
| **Perceived Severity** | | | | | |
| People with HPV might not have symptoms. | 4 (0.19) | 90 (4.18) | 1,481 (68.85) | 520 (24.17) | 56 (2.60) |
| HPV-associated warts could be uncomfortable or itchy. | 1 (0.05) | 42 (1.95) | 1,306 (60.72) | 739 (34.36) | 63 (2.93) |
| HPV-associated wart is a serious condition. | 1 (0.05) | 21 (0.98) | 1,245 (57.88) | 785 (36.49) | 99 (4.60) |
| HPV-associated cervical cancer is a serious condition. | 1 (0.05) | 9 (0.42) | 643 (29.89) | 1,183 (55.00) | 315 (14.64) |
| HPV-associated cervical cancer can occur in middle age. | 2 (0.09) | 23 (1.07) | 932 (43.33) | 1,062 (49.37) | 132 (6.14) |
| **Perceived Benefit** | | | | | |
| I trust vaccinations as it is getting better all the time because of research. | 3 (0.14) | 13 (0.60) | 480 (22.32) | 1,392 (64.71) | 263 (12.23) |
| The HPV vaccine is effective in preventing condyloma/genital warts in my daughter. | 0 (0.00) | 45 (2.09) | 1,170 (54.39) | 845 (39.28) | 91 (4.23) |
| HPV vaccine will be effective in preventing cervical cancer in my daughter. | 1 (0.05) | 22 (1.02) | 833 (38.73) | 1,166 (54.21) | 129 (6.00) |
| Vaccinated girls are less likely to get HPV than unvaccinated girls. | 0 (0.00) | 7 (0.33) | 647 (30.08) | 1,283 (59.65) | 214 (9.95) |
| HPV vaccination increases awareness of sexually transmitted diseases. | 0 (0.00) | 31 (1.44) | 864 (40.17) | 1,081 (50.26) | 175 (8.14) |
| **Perceived Barrier** | | | | | |
| I shall not vaccinate my daughter as it is painful. | 226 (10.51) | 1,451 (67.46) | 392 (18.22) | 79 (3.67) | 3 (0.14) |
| I shall not vaccinate my daughter as the HPV vaccine can cause an adverse effect. | 140 (6.51) | 1,179 (54.81) | 600 (27.89) | 219 (10.18) | 13 (0.60) |
| I shall not vaccinate my daughter as the HPV vaccine needs two injections. | 183 (8.51) | 1,296 (60.25) | 579 (26.92) | 89 (4.14) | 4 (0.19) |
| The HPV vaccine is so new that I want to wait a while before deciding if my daughter should get it. | 62 (2.88) | 537 (24.97) | 672 (31.24) | 794 (36.91) | 86 (4.00) |
| I shall vaccinate my daughter even if it is not free despite knowing that the HPV vaccine costs around 2500 BDT. | 69 (3.21) | 555 (25.80) | 722 (33.57) | 705 (32.78) | 100 (4.65) |

routine health checkups were more willing to vaccinate their children than others (p<0.001). Moreover, a considerable peak in vaccine acceptance level was observed among parents who heard about HPV (97.53%), HPV vaccine (97.71%), cervical cancer (93.33%), and cervical cancer vaccination (95.42%) compared to those who were unaware of that information (p<0.001). The vaccine acceptance level was positively associated with all the domains of the HBM except

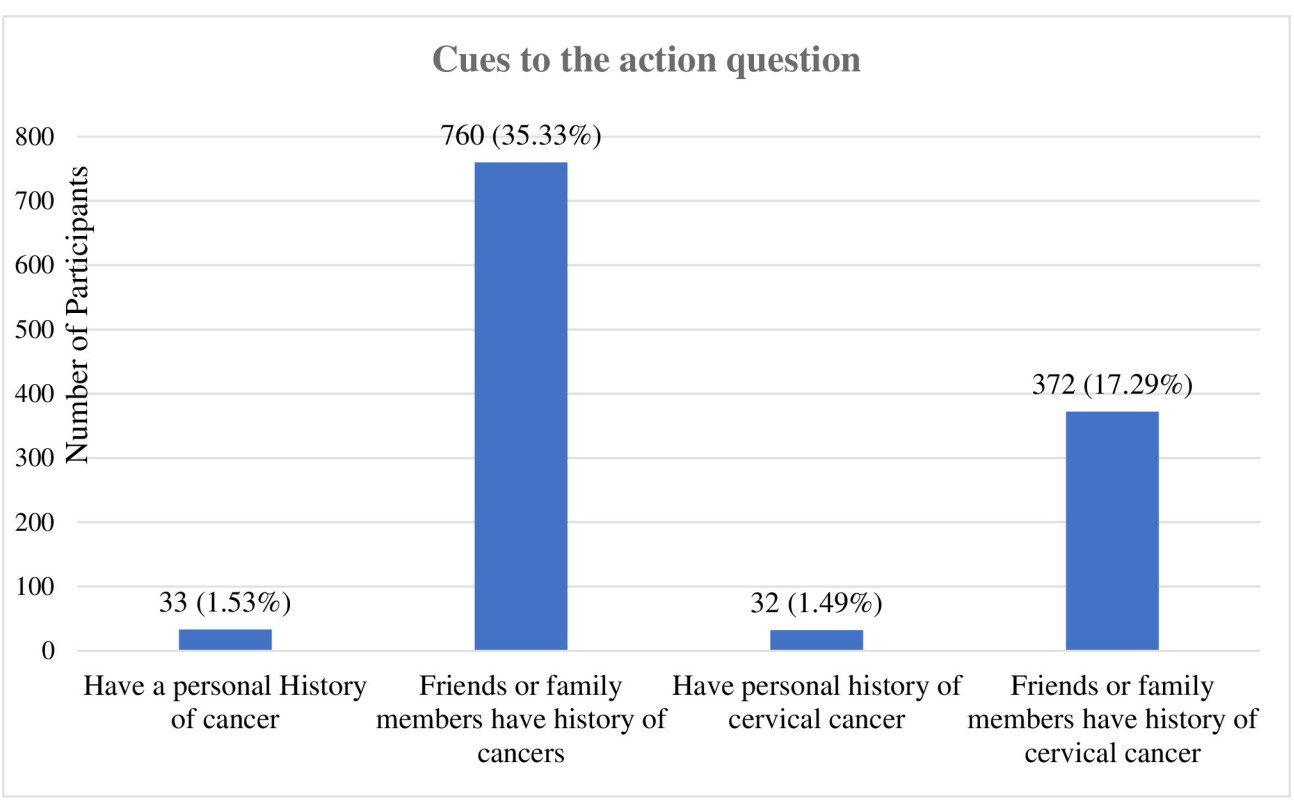

**Fig 2. Cues to the action question.**

in the perceived barrier domain, where no association was found (p = 0.489). Participants perceiving lesser vulnerability (97.63%) and lesser severity (97.60%) about the HPV, as well as greater benefits regarding the HPV vaccine (97.10%) and having apparent cues to actions (91.99%), demonstrated greater willingness to vaccinate their daughters.

To identify the factors influencing the acceptability of the HPV vaccine among parents of young daughters, both univariate and multivariate logistic regression analysis was carried out. (Table 4). After adjustment for other factors, it was found that a higher age was associated with a noticeable decrease in the parents' acceptance of vaccines for their daughters. (AOR: 0.92; 95% CI: 0.89–0.95). The urban residents exhibited 42% lower odds of vaccine acceptance than those in rural areas (AOR: 0.58; 95% CI: 0.36–0.92). Similarly, participants of the middle-income group had 44% lower odds of vaccination than the lower-income group. (AOR: 0.56; 95% CI: 0.32–0.97). Moreover, those with no history of regular health checkups had 45% lesser odds of vaccine acceptance than those with a history of regular health checkups. (AOR: 0.55; 95% CI: 0.35–0.85) Contrastingly, individuals with knowledge of cervical cancer had 4.10 times higher odds than those without such knowledge. (AOR: 4.10, 95%CI: 2.89–5.85). Concerning the perceived benefits, vaccine acceptance was 3.77-fold higher odds with positive attitudes towards the benefits of the HPV vaccine (AOR: 3.77, 95%CI: 2.32–6.35). Individuals experiencing cues to actions had 1.95 times higher odds of vaccinating their girls (AOR: 1.95, 95%CI: 1.37–2.80).

## Discussion

The acceptance of the Human Papillomavirus (HPV) vaccine has emerged as a critical determinant in the global effort to combat cervical cancer, a leading cause of morbidity and

**Table 3. Association of characteristics of the respondents with HPV vaccine acceptance.**

| Characteristic | Accepts Vaccine | | p-value |
|---|---|---|---|
| | No, (n = 288) | Yes, (n = 1,863) | |
| **Age (years)** | 40.70 ±6.20 | 37.79 ±5.71 | **<0.001**[2] |
| **Sex** | | | **<0.001**[3] |
| Female | 211 (12.05) | 1,540 (87.95) | |
| Male | 77 (19.25) | 323 (80.75) | |
| **Marital Status** | | | **0.027**[3] |
| Living with spouse | 263 (13.03) | 1,756 (86.97) | |
| Living without spouse | 18 (23.68) | 58 (76.32) | |
| Others | 7 (12.50) | 49 (87.50) | |
| **Religion** | | | 0.244[4] |
| Islam | 236 (13.23) | 1,548 (86.77) | |
| Hindu | 45 (13.35) | 292 (86.65) | |
| Buddhist | 0 (0.00) | 4 (100.00) | |
| Christian | 7 (26.92) | 19 (73.08) | |
| **Residence** | | | **0.018**[3] |
| Rural | 89 (11.37) | 694 (88.63) | |
| Semi-urban | 28 (10.85) | 230 (89.15) | |
| Urban | 171 (15.41) | 939 (84.59) | |
| **Years of Education** | 10.09 ±4.06 | 11.57 ±4.54 | **<0.001**[2] |
| **Occupation** | | | **<0.001**[3] |
| Job | 28 (5.48) | 483 (94.52) | |
| Business | 55 (29.57) | 131 (70.43) | |
| Housewife | 191 (14.00) | 1,173 (86.00) | |
| Others | 14 (15.56) | 76 (84.44) | |
| **Health Care Worker** | 0 (0.00) | 183 (100.00) | **<0.001**[3] |
| **Monthly household income (BDT)** | | | **0.003**[3] |
| < = 20000 | 63 (11.52) | 484 (88.48) | |
| 20001–35000 | 63 (11.07) | 506 (88.93) | |
| 35001–50000 | 106 (17.64) | 495 (82.36) | |
| >50000 | 55 (12.76) | 376 (87.24) | |
| **Family Type** | | | **<0.001**[3] |
| Nuclear | 89 (9.65) | 833 (90.35) | |
| Joint | 199 (16.21) | 1,029 (83.79) | |
| **Routine Health Checkup** | | | **<0.001**[3] |
| Regular | 41 (6.11) | 630 (93.89) | |
| Irregular | 160 (17.98) | 730 (82.02) | |
| Never | 87 (14.75) | 503 (85.25) | |
| **Heard about Human Papilloma Virus** | 12 (2.47) | 473 (97.53) | **<0.001**[3] |
| **Heard about HPV Vaccine** | 11 (2.29) | 469 (97.71) | **<0.001**[3] |
| **Heard about Cervical Cancer** | 115 (6.67) | 1,609 (93.33) | **<0.001**[3] |
| **Heard about Cervical Cancer Vaccination** | 48 (4.58) | 1,000 (95.42) | **<0.001**[3] |
| **Perceived Vulnerability** | | | **<0.001**[3] |
| Negative | 277 (16.42) | 1,410 (83.58) | |
| Positive | 11 (2.37) | 453 (97.63) | |
| **Perceived Severity** | | | **<0.001**[3] |
| Negative | 278 (16.03) | 1,456 (83.97) | |

*(Continued)*

**Table 3.** (Continued)

| Characteristic | Accepts Vaccine | | p-value |
| --- | --- | --- | --- |
| | No, (n = 288) | Yes, (n = 1,863) | |
| Positive | 10 (2.40) | 407 (97.60) | |
| **Perceived Benefit** | | | **<0.001**[3] |
| Negative | 264 (19.94) | 1,060 (80.06) | |
| Positive | 24 (2.90) | 803 (97.10) | |
| **Perceived Barrier** | | | 0.489[4] |
| Negative | 285 (13.36) | 1,849 (86.64) | |
| Positive | 3 (17.65) | 14 (82.35) | |
| **Cues to Action** | | | **<0.001**[3] |
| Negative | 216 (17.25) | 1,036 (82.75) | |
| Positive | 72 (8.01) | 827 (91.99) | |

[1]Mean ±SD; n (%); Median (IQR)

[2]Welch Two Sample t-test

[3]Pearson's Chi-squared test

[4]Fisher's exact test

[5]Wilcoxon rank sum test

Significant p-values are shown in bold

mortality among women [27]. In the context of Bangladesh, where cervical cancer is the second most common cancer among females and poses a significant health burden [28], understanding the factors that influence parental acceptance of the HPV vaccine is of paramount importance. This nationwide study employs the Health Belief Model as a framework to explore and elucidate the multifaceted factors that shape HPV vaccine acceptance among Bangladeshi parents.

Our study observed an impressively high acceptance rate of the HPV vaccine among parents, with an overall acceptance rate of 86.61%. This finding demonstrates a substantial willingness among Bangladeshi parents to embrace HPV vaccination as a crucial preventive measure against cervical cancer. It's important to note the variability in global acceptance of the HPV vaccine, as approval rates in different studies range from 44% to 79% [29–34]. Notably, two studies conducted in Japan and Hong Kong found that less than a third of mothers expressed the intention to vaccinate their daughters [35, 36]. When compared to prior research in similar contexts in India and China, where parental acceptance of the HPV vaccine was approximately 71% and 83%, respectively [13, 19], our study stands out with its significantly higher acceptance rate. However, compared to other countries, recent nationwide vaccination for COVID-19 might have been a mover for higher acceptance of the vaccine in general, explaining the findings in our study.

The sociodemographic analysis reveals pivotal factors influencing HPV vaccine acceptance among Bangladeshi parents. Notably, younger parents show a greater propensity for vaccine acceptance, aligning with previous studies conducted in comparable contexts in Iran and Ethiopia [37, 38]. This inclination is likely attributed to their greater exposure to health-related information and receptiveness to novel healthcare interventions. The acceptance rate is notably influenced by sex, as females exhibit a higher acceptance rate than males, which mirrors consistent trends detected in research conducted in various geographic areas [39, 40]. This variation can be attributed to the traditional caregiving roles of females in healthcare decision-

**Table 4. Determinants of HPV vaccine acceptance.**

| Characteristic | Univariate Models | | | Multivariate Model | | |
| --- | --- | --- | --- | --- | --- | --- |
| | OR | 95% CI | p-value | OR | 95% CI | p-value |
| **Age (years)** | 0.92 | 0.90 to 0.94 | **<0.001** | 0.92 | 0.89 to 0.95 | **<0.001** |
| **Sex** | | | | | | |
| Female | — | — | | — | — | |
| Male | 0.57 | 0.43 to 0.77 | **<0.001** | 1.05 | 0.53 to 2.04 | 0.882 |
| **Marital Status** | | | | | | |
| Living with spouse | — | — | | — | — | |
| Living without spouse | 0.48 | 0.29 to 0.85 | **0.009** | 0.72 | 0.37 to 1.46 | 0.349 |
| Others | 1.05 | 0.50 to 2.56 | 0.908 | 0.93 | 0.38 to 2.56 | 0.886 |
| **Residence** | | | | | | |
| Rural | — | — | | — | — | |
| Semi-urban | 1.05 | 0.68 to 1.68 | 0.821 | 0.72 | 0.42 to 1.27 | 0.248 |
| Urban | 0.70 | 0.53 to 0.92 | **0.012** | 0.58 | 0.36 to 0.92 | **0.021** |
| **Years of Education** | 1.07 | 1.05 to 1.10 | **<0.001** | 1.02 | 0.97 to 1.08 | 0.375 |
| **Occupation** | | | | | | |
| Job | — | — | | — | — | |
| Business | 0.14 | 0.08 to 0.22 | **<0.001** | 0.59 | 0.31 to 1.10 | 0.098 |
| Housewife | 0.36 | 0.23 to 0.53 | **<0.001** | 0.62 | 0.33 to 1.11 | 0.118 |
| Others | 0.31 | 0.16 to 0.64 | **<0.001** | 0.63 | 0.27 to 1.53 | 0.300 |
| **Monthly household income (BDT)** | | | | | | |
| < = 20000 | — | — | | — | — | |
| 20001–35000 | 1.05 | 0.72 to 1.52 | 0.814 | 0.85 | 0.54 to 1.35 | 0.504 |
| 35001–50000 | 0.61 | 0.43 to 0.85 | **0.004** | 0.56 | 0.32 to 0.97 | **0.039** |
| >50000 | 0.89 | 0.61 to 1.31 | 0.553 | 0.57 | 0.30 to 1.08 | 0.084 |
| **Family Type** | | | | | | |
| Nuclear | — | — | | — | — | |
| Joint | 0.55 | 0.42 to 0.72 | **<0.001** | 0.79 | 0.56 to 1.11 | 0.171 |
| **Routine Health Checkup** | | | | | | |
| Regular | — | — | | — | — | |
| Irregular | 0.30 | 0.20 to 0.42 | **<0.001** | 0.70 | 0.46 to 1.07 | 0.105 |
| Never | 0.38 | 0.25 to 0.55 | **<0.001** | 0.55 | 0.35 to 0.85 | **0.007** |
| **Heard about Human Papilloma Virus** | | | | | | |
| No | — | — | | — | — | |
| Yes | 7.83 | 4.55 to 14.9 | **<0.001** | 1.16 | 0.42 to 3.49 | 0.783 |
| **Heard about HPV Vaccine** | | | | | | |
| No | — | — | | — | — | |
| Yes | 8.47 | 4.83 to 16.6 | **<0.001** | 1.45 | 0.49 to 4.43 | 0.515 |
| **Heard about Cervical Cancer** | | | | | | |
| No | — | — | | — | — | |
| Yes | 9.53 | 7.28 to 12.5 | **<0.001** | 4.10 | 2.89 to 5.85 | **<0.001** |
| **Heard about Cervical Cancer Vaccination** | | | | | | |
| No | — | — | | — | — | |
| Yes | 5.79 | 4.23 to 8.09 | **<0.001** | 1.32 | 0.86 to 2.05 | 0.211 |
| **Perceived Vulnerability** | | | | | | |
| Negative | — | — | | — | — | |
| Positive | 8.09 | 4.61 to 15.8 | **<0.001** | 1.77 | 0.89 to 3.87 | 0.125 |
| **Perceived Severity** | | | | | | |

*(Continued)*

**Table 4.** (Continued)

| Characteristic | Univariate Models | | | Multivariate Model | | |
|---|---|---|---|---|---|---|
| | OR | 95% CI | p-value | OR | 95% CI | p-value |
| Negative | — | — | | — | — | |
| Positive | 7.77 | 4.32 to 15.8 | **<0.001** | 1.29 | 0.62 to 2.93 | 0.523 |
| **Perceived Benefit** | | | | | | |
| Negative | — | — | | — | — | |
| Positive | 8.33 | 5.55 to 13.1 | **<0.001** | 3.77 | 2.32 to 6.35 | **<0.001** |
| **Perceived Barrier** | | | | | | |
| Negative | — | — | | — | — | |
| Positive | 0.72 | 0.23 to 3.13 | 0.606 | 0.32 | 0.07 to 1.87 | 0.166 |
| **Cues to Action** | | | | | | |
| Negative | — | — | | — | — | |
| Positive | 2.39 | 1.82 to 3.19 | **<0.001** | 1.95 | 1.37 to 2.80 | **<0.001** |

OR = Odds Ratio, CI = Confidence Interval; Dashes (-) represent reference values. Significant p-values are shown in bold.

making. However, it's worth noting that the percentage of males who participated in our study was 18.60%, indicating a relatively small sample size.

A significant observation in our study was that both urban residents and individuals in the middle-income group had lower vaccine acceptance rates compared to their counterparts in rural areas and lower-income groups, respectively. This aligned with research by Liu et al. [41], which revealed that individuals who were well-educated, had higher incomes, and lived in urban areas exhibited greater vaccine hesitancy. Similarly, Wagner et al. [42] and Lin et al. [43] reported similar results, indicating lower vaccine hesitancy and fewer safety concerns in low and middle-income regions compared to high-income areas of China. The higher acceptance in rural communities of Bangladesh with relatively low economic capacity could be explained by the fact that healthcare workers employed by the Ministry of Health, including health assistants and community healthcare providers, play a proactive role in increasing vaccine uptake in rural areas, while the urban health care services managed by the city corporation lacks such community-centric health delivery system.

The study revealed a strong correlation between regular health checkups and an enhanced inclination to receive vaccinations, affirming prior research by Wheldon et al. that consistently underscored the enduring importance of readily accessible healthcare in influencing vaccination behavior [44]. Furthermore, a systematic review and meta-analysis conducted by Newman et al. identified a positive association between routine child preventive check-ups and parents' willingness to accept the HPV vaccine [45]. Individuals who undergo regular health check-ups are more likely to be self-conscious about their health compared to those who do not have regular health check-ups, explaining the higher HPV vaccine acceptance among the former.

Furthermore, in the analysis, it became evident that having knowledge about cervical cancer and HPV significantly motivated vaccine acceptance. This aligned with previous research by Zouheir et al. (2016) [46], which highlighted the impact of these factors as influential. A similar trend was noted in a study by Zhou et al. (2019) [47], where parental awareness was a positive factor influencing the intention for HPV vaccination among adolescents. This emphasizes the vital need for specific awareness initiatives targeting these sociodemographic aspects to enhance the uptake of the HPV vaccine.

The application of the Health Belief Model (HBM) to understand vaccine acceptance in this context yielded several important insights. Participants generally recognized that young women were susceptible to HPV and were worried about their children getting infected with the virus, showing a sense of vulnerability to it. Nevertheless, there was a degree of uncertainty regarding the seriousness of HPV, especially concerning the presence of symptoms. This finding corresponds with earlier research by Vermandere et al. and Yarici et al. [48, 49], which highlighted differing levels of comprehension regarding the health risks linked to HPV infection due to limited knowledge and awareness. Nevertheless, the majority of our participants had trust in vaccinations and believed the HPV vaccine would effectively prevent cervical cancer in their daughters. This perception of benefit emerged as an independent determinant of HPV vaccine acceptance in the multivariable model. Suo et al. [50] also observed that the perceived safety of the HPV vaccine improved its acceptance among parents of adolescent girls. Participants mostly disagreed with potential barriers to HPV vaccination, such as concerns about pain, side effects, and the number of shots required. Similar findings were reported in studies by Guvenc et al. and Marlow et al. [51, 52], indicating that these barriers typically do not pose significant obstacles to vaccine acceptance. Individuals often prioritize the perceived benefits and risks associated with HPV vaccination over the potential barriers. We found that even perceived severity was superseded by the perception of benefit in vaccine acceptance. Additionally, our multivariable analysis revealed that cues to action was another independent determinant of vaccine acceptance. Individuals who had history of any cervical cancer or other cancer, and individual who saw any family member or friend affected by those cancers were significantly most likely to accept the vaccine, because they had first-hand experience of the consequences of cancers.

The Global Vaccine Action Plan aims to save lives by ensuring equitable vaccine access worldwide [53]. In Bangladesh, the battle against cervical cancer is paramount. However, the findings of this nationwide study revealed that HPV vaccine acceptance is shaped by factors like age, maternal engagement, and healthcare access. To address this, tailored awareness campaigns, educational initiatives, and enhanced healthcare services are imperative. Overcoming barriers and implementing effective cues can significantly enhance vaccine acceptance, a crucial step in the fight against cervical cancer.

## Strengths and limitations

The strengths of this study are grounded in its nationwide coverage, offering a comprehensive understanding of HPV vaccine acceptance across diverse regions of Bangladesh. Furthermore, the structured application of the Health Belief Model as a framework allows for a systematic and theory-driven analysis of the factors influencing vaccine acceptance. Additionally, the study underwent a rigorous piloting process, which contributed to the refinement of survey instruments, ensuring the clarity and consistency of data collection. Moreover, the study also considered a wide range of sociodemographic variables, adding depth to the analysis and contributing to a more nuanced understanding of the determinants of vaccine acceptance. These collective strengths bolster the robustness, representativeness, and methodological rigor of the study's findings, providing valuable insights into the landscape of HPV vaccine acceptance in Bangladesh.

The limitations of the study include its cross-sectional design, providing a brief glimpse into HPV vaccine acceptance without the ability to establish causation or observe changes over time. The use of convenience sampling may introduce biases affecting the study's overall representation. Another limitation is that the representation of fathers was low in our study, potentially missing their intentions regarding HPV vaccine acceptance for their daughters.

The lack of longitudinal data hinders the exploration of acceptance trends. Additionally, the study predominantly offers quantitative insights, with limited qualitative depth, to probe into the reasons behind the socio-demographic determinants of HPV vaccine acceptance. Furthermore, the study did not thoroughly investigate external influences on acceptance, such as policy shifts or external awareness campaigns, which could have influenced attitudes and behaviours. These aspects should be taken into account when interpreting the study's outcomes.

## Conclusion

Our nationwide study found a reasonably good level of acceptance of the HPV vaccine among the parents of eligible daughters. The vaccine acceptance level was positively associated with almost all the domains of the HBM. Multiple factors such as younger age, urban residence, monthly income in the lowest quartile, history of regular routine health check-ups, knowledge of cervical cancer, positive perception about vaccine benefits, and positive cues to actions were associated with HPV vaccine acceptance. However, as the awareness level is still very low, the government should take steps to initiate awareness campaigns at all levels.

## Recommendation

- Implement tailored awareness campaigns for specific demographic groups, including younger parents, mothers, urban residents, and middle-income individuals.

- Implement policies and interventions to improve healthcare availability, particularly in urban areas and among the middle-income population, ensuring that parents have easy access to vaccination services.

- Strengthen educational initiatives to increase awareness about cervical cancer and the benefits of HPV vaccination.

- Conduct longitudinal studies to track trends in HPV vaccine acceptance over time and assess the long-term effectiveness of interventions aimed at increasing vaccine uptake.

## Supporting information

**S1 Table. The division-wise distribution of samples.**
(DOCX)

**S1 File. Completed questionnaire used for data collection.**
(DOCX)

**S1 Dataset. Dataset used to generate figures, tables, and statistics.**
(CSV)

## Acknowledgments

The authors would like to extend their profound gratitude to all the participants in this study, who voluntarily and spontaneously contributed to our research.

## Author Contributions

**Conceptualization:** Mohammad Delwer Hossain Hawlader, Fahima Nasrin Eva.

**Data curation:** Md. Abdullah Saeed Khan, Tariful Islam, Nur-E-Safa Meem.

**Formal analysis:** Md. Abdullah Saeed Khan.

**Investigation:** Mohammad Delwer Hossain Hawlader, Fahima Nasrin Eva, Tariful Islam, Umme Kulsum Monisha, Irin Chowdhury, Nur-E-Safa Meem, Mohammad Ali Hossain, Arpita Goutam, Tahmina Zerin, Nishat Alam, Rima Nath, Shamma Sifat, Sayla Sultana, Mosammat Sadeka Sultana, Sumit Kumar Saha, Naifa Enam Sarker.

**Methodology:** Fahima Nasrin Eva, Md. Abdullah Saeed Khan, Irin Chowdhury, Rifat Ara, Arpita Goutam, Mohammad Lutfor Rahman.

**Project administration:** Mohammad Delwer Hossain Hawlader, Fahima Nasrin Eva, Md. Abdullah Saeed Khan.

**Resources:** Mohammad Delwer Hossain Hawlader, Fahima Nasrin Eva, Tariful Islam, Umme Kulsum Monisha, Irin Chowdhury, Nur-E-Safa Meem, Mohammad Ali Hossain, Arpita Goutam, Tahmina Zerin, Nishat Alam, Rima Nath, Shamma Sifat, Sayla Sultana, Mosammat Sadeka Sultana, Sumit Kumar Saha, Naifa Enam Sarker.

**Software:** Md. Abdullah Saeed Khan, Tariful Islam, Nur-E-Safa Meem.

**Supervision:** Mohammad Delwer Hossain Hawlader, Mohammad Hayatun Nabi, Mohammad Lutfor Rahman.

**Validation:** Mohammad Delwer Hossain Hawlader, Fahima Nasrin Eva, Tariful Islam, Umme Kulsum Monisha, Irin Chowdhury, Rifat Ara, Nur-E-Safa Meem, Mohammad Ali Hossain, Arpita Goutam, Tahmina Zerin, Nishat Alam, Rima Nath, Shamma Sifat, Sayla Sultana, Mosammat Sadeka Sultana, Sumit Kumar Saha, Naifa Enam Sarker.

**Visualization:** Rifat Ara, Mohammad Hayatun Nabi, Mohammad Lutfor Rahman.

**Writing – original draft:** Fahima Nasrin Eva, Umme Kulsum Monisha, Irin Chowdhury, Rifat Ara, Mohammad Ali Hossain, Arpita Goutam.

**Writing – review & editing:** Mohammad Delwer Hossain Hawlader, Md. Abdullah Saeed Khan, Mohammad Hayatun Nabi, Mohammad Lutfor Rahman.

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
