## [Decision Letter · Decision Letter 0]

9 Jan 2024

PONE-D-23-37347Acceptance of Human Papillomavirus (HPV) vaccine among the parents of eligible daughters (9-15 years) in Bangladesh: a nationwide study using Health Belief ModelPLOS ONE

Dear Dr. Eva,

Thank you for submitting your manuscript to PLOS ONE. After careful consideration, we feel that it has merit but does not fully meet PLOS ONE’s publication criteria as it currently stands. Therefore, we invite you to submit a revised version of the manuscript that addresses the points raised during the review process.

We look forward to receiving your revised manuscript.

Kind regards,

Kehinde Kazeem Kanmodi, BDS

Academic Editor

PLOS ONE

Journal Requirements:

Additional Editor Comments:

Nil.

Reviewers' comments:

Reviewer's Responses to Questions

**Comments to the Author**

1. Is the manuscript technically sound, and do the data support the conclusions?

Reviewer #1: Yes

Reviewer #2: Yes

Reviewer #3: Partly

Reviewer #4: Yes

2. Has the statistical analysis been performed appropriately and rigorously? 

Reviewer #1: Yes

Reviewer #2: Yes

Reviewer #3: Yes

Reviewer #4: I Don't Know

3. Have the authors made all data underlying the findings in their manuscript fully available?

Reviewer #1: Yes

Reviewer #2: Yes

Reviewer #3: No

Reviewer #4: No

4. Is the manuscript presented in an intelligible fashion and written in standard English?

Reviewer #1: Yes

Reviewer #2: Yes

Reviewer #3: Yes

Reviewer #4: Yes

5. Review Comments to the Author

Reviewer #1: Few minor changes are required . Which being mentioned in the uploaded script. It may be considered for publication following the minor changes are made. The ethical permission part needs to be considered and Bangladesh bio-ethics society permission needs to be considered. Recommendations needs to be added in bullet points . It will provide the implication of this study.

Reviewer #2: Thank you Authors for coming up with this interesting manuscript. However, I have made the following review comments for better quality of the manuscript.

1. Under the methodology, authors should endeavor to shed more light on the content validity of the questionnaire.

Also, the total adjusted sample size was 2160 and the total sample size used after exclusion was 2151. Can authors expatiate on this disparity?. Again, this is a national survey, is the sample size representative?

The pretest and pilot protocols were not referenced. Authors should kindly reference this for better understanding of the rationale behind the pretest and pilot protocol employed.

2. Authors should evaluate the manuscript for grammatical errors overall. For example, Lines 295-296 do not read well. ....."Maternal gender emerges as significant determinant, with mothers demonstrating higher acceptance rate than fathers...." This should be corrected.

Overall, I will suggest a minor revision to this manuscript.

Reviewer #3: I thank the Editor for inviting me to review this important manuscript. The authors have done a very good job of performing this needful study. The study has several policy implications and strengths, including large sample size, unprecedentedness, and rigor. However, I still believe the manuscript needs substantial improvements in the ways it is prepared by the authors. My comments to them are:

1. Lines 65 and 66, without reference, the authors wrote: “In high-income countries, the coverage is almost 80%.”

2. Line 83, the authors wrote “mental illnesses were excluded.”

3. Lines 83-84, the authors wrote: “To ensure representative sampling, we determined division 84 specific sample sizes using the 2022 Population & Housing Census (13)”. Firstly, this statement about the study being “representative” is not objective. Second, the reference you provided opens to a page I assume to be in Bengal language. To make it easy for the authors and the reader to ascertain the study’s so-called representativeness, the authors should consider to extract the socio-demographics of Bangladeshi population from this page or any other reference and create a table showing the study’s socio-demographics on one hand and the Bangladeshi general population on the other [e.g. study’s female N 9%) vs national female N (%), study’s rural N (%) vs national rural dwellers N (%), e.t.c.]. In this way, it will be known objectively whether the study is entirely, partially, or completely not representative of the Bangladeshi population.

4. Lines 85-86: without providing the reference, the authors wrote “while considering an 80% vaccine acceptance among parents based on the existing literature”

5. Lines 88-89: the authors wrote “of Bangladesh between June 28 and August 2, 2023, resulting in a total sample size of 2,151 after exclusions”. Your reader may be interested in knowing the overall response rate when invitation was issued to potential participants, total responses before the exclusion, and the exclusion criteria.

6. Also, can the authors consider attaching an English version of their study questionnaire as supplement?

7. Lines 117-120, why is this statement (Second, a pre-test was conducted among 28 parents of 118 daughters aged between 9–15 years old prior to the start of the actual data collection. In order to make the questionnaire more understandable for the participants and to improve face- and construct validity, some questions were rewritten or clarified.) bolder than the rest of the manuscript?

8. Pages 4, 5, and 6 have some redundancy of information. Please, try to scale down this. You may which to just reference the study questionnaire you attach in the supplement.

9. How was knowledge scored?

10. Also try to disaggregate the references for each questionnaire used to assess the various domains of the study, e.g. knowledge(reference), HBM(reference)

11. On what basis did the authors classify monthly household income? One would expect the country’s minimum wage to be the reference for the income groups. Also, in no where in your manuscript did you spelt out your “BDT”

12. Line 136, The authors also mentioned “Academic institution type (government, private, madrasa)” as part of the study’s socio-demographic variables, which is important. However, they have not included this in the analysis without any explanation for dropping it.

13. In the regression model, why have the authors not considered exploring the eight divisions of the country as potential determinant?

14. There’s plethora of evidence supporting a parent’s/caregiver’s “level” of education with vaccination. However, I noticed the authors used “years of education” instead of “levels (primary, secondary, tertiary, university, e.t.c.”), which is more objective and informative. Why have the authors used their approach? I ask this because years of education does not necessarily mean level of education since one, for example, can decide to continue doing certificates after a primary or secondary education without progressing to a university degree.

15. Lines 287-290 in the discussion, the authors wrote: “When compared to prior research in similar contexts in India and China, where parental acceptance of the HPV vaccine was approximately 71% and 83%, respectively (11,29), our study stands out with its significantly higher acceptance rate. This indicates a promising upward trend in HPV vaccine acceptance within Bangladesh.” Why would you cite studies from India and China that reported an acceptance rate lower than that of your study and then say” This indicates a promising upward trend in HPV vaccine acceptance “within” Bangladesh. For you to say “upward trend within Bangladesh’, the reference studies have to be from Bangladesh as well.

16. In line 295, the authors wrote: “The maternal gender emerges…..”. What is the meaning of this phrase: and why are you now mentioning gender after you’ve explicitly mentioned in the Methods and Results that you evaluated “sex”. And why maternal? Or are you saying the women you evaluated are pregnant or what?

17. Lines 377 and 378 of the Conclusion, the authors wrote: “Our nationwide study found a very high level of acceptance of the HPV vaccine among the parents of eligible daughters”, referring to the study’s acceptance rate of 86%. If we assume that, all the 86% who accepted actually have their daughters vaccinated, will this proportion meet of with the WHO target of 2030 you cited in your introduction?

18. In the Conclusion, lines 379-381, the authors wrote: “Multiple factors such as age, residence, education, income, history of medical check-ups, knowledge of cervical cancer, having a positive attitude towards the benefits, and cues to actions were associated with HPV vaccine acceptance.” This statement is not supported by the study findings because some of the variables mentioned were not statistically significant at the multivariate level of analysis done by the authors. Consider to revise specific to your findings.

I look forward to reading the revised version of this manuscript

Reviewer #4: Comment to authors

Title: Acceptance of Human Papillomavirus (HPV) vaccine among the parents of eligible daughters (9-15 years) in Bangladesh: a nationwide study using Health Belief Model

Manuscript number: PONE-D-23-37347

Abstract

Background – how parent acceptance contribute to ensure 90% of girls receive the HPV vaccine before turning 15?

Method – make it summary of method section. For instance, how was sampling conducted? How data were collected?

Result – provide the finding with its upper and lower limit of the prevalence.

Conclusion – what is authors’ landmark to reach at high acceptance? “The vaccine acceptance was positively associated with almost all the domains of the HBM.” What authors want to convey? If it is to indicate direction of association; all their factors are preventive.

Line 45-46: “Multiple factors such as age, residence, education, income, history of medical check-ups, knowledge of cervical cancer, and cues to actions were associated with HPV vaccine acceptance” These are crude to be used in conclusion; be specific, which classification are associated with outcome? For instance is it younger age or older? Is it having higher degree or not attending formal education?

Introduction

How many of the increased coverage of HPV vaccine is attributable to HPV vaccine acceptance of parent across the globe or in low and middle income countries? What is really matter? What are role of knowledge and attitude of the parents? What factors were contribute to high or low acceptance of the vaccine when you summarize previous studies? How you intend to uncover their gaps?

Method

How authors maintained external validity of the finding? “We collected samples conveniently from all eight divisions” How conveniently sampled study was generalized to source?

Line 99-100; how you minimize interviewer bias?

Authors should explain how they measure HPV vaccine acceptance well. The measurement for all potential variables were not described? Authors should also provide appropriate citations for measurement of outcome and explanatory variables anywhere appropriate.

How variables were selected for multivariable analysis? How model fitness were check? Was there multi-collinearity?

Result

Line 186: No sentence or paragraph should begin with number.

Table 1 and 2: what is n? Is it representing sample or frequency? Check and correct. Indentation needed for classification of variables to make easy for readers.

Line 230-250: the explanation for crude analysis are not needed. There are a lot of confounding for these association, hence you need to explain after controlling for these confounding.

Can authors merge findings of table 3 and 4?

How many variables can be included in the model at once? In this study about 19 variables were run at once!

Table 4- what are dashes (-) represent? How about bolds?

Discussion

Line 274-280: the discussion usually begin with purpose of the study and summary of findings.

Why acceptance of the vaccine in Bangladesh were higher compared to other countries like china and Japan?

Authors should explain why those in urban and middle income are lower acceptance compared to rural and low income in contradiction to previous findings.

Line 311-318: why those who have no routine checkup have lower odds of acceptance? What really contributed to this association?

Line 326-348: what are the relationship between HBM and acceptance? In your table 4, only perceived benefit and cue to action are associated. Focus on you finding and explain why and its implications.

Conclusion

The same comments to conclusion in abstract.

Why Fig 1 is three dimensional? Is key is not acceptance rather willing to vaccine, do you think they are the same? Is accepting is willing to vaccinate?

Figure 2, what is x-y axis indicate? Label them. Figure title should fulfil its criteria.

6. PLOS authors have the option to publish the peer review history of their article (what does this mean?). If published, this will include your full peer review and any attached files.

Reviewer #1: No

Reviewer #2: No

Reviewer #3: **Yes: **Sahabi Kabir Sulaiman

Reviewer #4: **Yes: **Kasiye Shiferaw

---

## [Author Response · Author response to Decision Letter 0]

1 Apr 2024

Response to Reviewer#1’s Comment: 

Comment: Few minor changes are required. Which being mentioned in the uploaded script. It may be considered for publication following the minor changes are made. The ethical permission part needs to be considered and Bangladesh bio-ethics society permission needs to be considered. Recommendations needs to be added in bullet points. It will provide the implication of this study.

Response: Thank you for your valuable feedback. Our study has received IRB approval (Approval no: 2023/OR-NSU/IRB/0507) from North South University’s Ethical Review Committee. Additionally, we have followed the ethical standards outlined in the 1964 Helsinki Declaration and its subsequent amendments wherever applicable and the STROBE guidelines to craft the manuscript (Kindly refer to the lines 193-201 in the revised manuscript). We believe this oversight adequately addresses ethical considerations. 

Moreover, we have incorporated the recommendations in bullet points in the revised manuscript in the recommendation section (Please refer to lines 408-415) to highlight the implications of our study.

Response to Reviewer#2’s Comment: 

Comment: Thank you Authors for coming up with this interesting manuscript. However, I have made the following review comments for better quality of the manuscript.

1. Under the methodology, authors should endeavor to shed more light on the content validity of the questionnaire.

Also, the total adjusted sample size was 2160 and the total sample size used after exclusion was 2151. Can authors expatiate on this disparity? Again, this is a national survey, is the sample size representative?

The pretest and pilot protocols were not referenced. Authors should kindly reference this for better understanding of the rationale behind the pretest and pilot protocol employed.

Response: Thank you for your comment. The questionnaire underwent a rigorous validation process and was reviewed by study supervisors and expert researchers, ensuring its appropriateness and relevance to the study objectives. Content validity was addressed through pre-testing, construct definition, item analysis, modification, pilot study and continuous review, as detailed in the questionnaire part (Please refer to lines 117-141 in the revised manuscript) under the methodology section. We believe these steps robustly address the content validity concerns of the questionnaire.

In response to the reviewer's query regarding the disparity between the total adjusted sample size (2160) and the total sample size used after exclusion (2151), we would like to clarify that the difference arises from the exclusion of participants who had incomplete data. These exclusions were necessary to ensure the integrity and validity of the study results.

In response to the concern about the representativeness of the sample size in our nationwide study, we want to highlight that we meticulously covered all eight divisions of Bangladesh and collected samples from 42 out of 64 districts. This extensive geographical coverage enhances the representativeness of our sample, ensuring a comprehensive reflection of diverse demographic and regional characteristics across the country. We have provided detailed information about the sampling strategy and coverage in the methodology section (Please refer to lines 91-116 in the revised manuscript) of the manuscript.

To address the reviewer's comment regarding the referencing of pretest and pilot protocols, we appreciate the suggestion. However, the detailed protocols for the pretest and pilot study were not published separately as part of the manuscript. Hence, direct referencing to specific pilot protocols is not feasible. To enhance clarity, we have included relevant information about the pretest and pilot study within the questionnaire section (Lines 133-138) of the methodology, providing a comprehensive understanding of the rationale and implementation. We trust this clarification meets the reviewer's expectations.

Comment: 2. Authors should evaluate the manuscript for grammatical errors overall. For example, Lines 295-296 do not read well. ....."Maternal gender emerges as significant determinant, with mothers demonstrating higher acceptance rate than fathers...." This should be corrected.

Overall, I will suggest a minor revision to this manuscript.

Response: Thank you for your valuable feedback. We have carefully reviewed and revised the manuscript for grammatical errors, including the mentioned lines. The line is now read as follows (Lines 318-320: The acceptance rate is notably influenced by sex, as females exhibit a higher acceptance rate than males….). We believe this adjustment improves the clarity and readability of the manuscript.

Response to Reviewer#3’s Comment: 

Comment: I thank the Editor for inviting me to review this important manuscript. The authors have done a very good job of performing this needful study. The study has several policy implications and strengths, including large sample size, unprecedentedness, and rigor. However, I still believe the manuscript needs substantial improvements in the ways it is prepared by the authors. My comments to them are:

Comment: 1. Lines 65 and 66, without reference, the authors wrote: “In high-income countries, the coverage is almost 80%.”

Response: Thank you for bringing this to our attention. We have now appropriately referenced the statement in Lines 65-66. The relevant citations have been added to ensure accuracy and credibility. Please refer to the updated manuscript for the correct referencing (Lines 67-68).

Comment: 2. Line 83, the authors wrote “mental illnesses were excluded.”

Response: Thank you for pointing this out. We agree that the statement 'mental illnesses were excluded' can be clarified for better understanding. In the revised manuscript, we have provided a more detailed explanation of the criteria and process used for the exclusion of mental illnesses. Please refer to lines 93-94 in the revised manuscript, which is now read as follows-

“…..those who were diagnosed and were taking medication for mental health illness were excluded.”

Comment: 3. Lines 83-84, the authors wrote: “To ensure representative sampling, we determined division 84 specific sample sizes using the 2022 Population & Housing Census (13)”. Firstly, this statement about the study being “representative” is not objective. Second, the reference you provided opens to a page I assume to be in Bengal language. To make it easy for the authors and the reader to ascertain the study’s so-called representativeness, the authors should consider to extract the socio-demographics of Bangladeshi population from this page or any other reference and create a table showing the study’s socio-demographics on one hand and the Bangladeshi general population on the other [e.g. study’s female N 9%) vs national female N (%), study’s rural N (%) vs national rural dwellers N (%), e.t.c.]. In this way, it will be known objectively whether the study is entirely, partially, or completely not representative of the Bangladeshi population.

Response: Thank you very much for the suggestion. The reference you mentioned (reference no 13, now 18) opens the preliminary report of the Population and Housing Census 2022. Please note that the number ‘84’ in the sentence that you copied is the line number. It does not refer to anything else. Note also that our target was to ensure proportionate representation of the eight administrative divisions of Bangladesh. Based on your suggestion, we have included a supplementary table (S1 Table) where the population of each division with their proportion is listed alongside a number of samples taken from each division with their proportion. We extracted this data from the preliminary reports of the population and housing census available for free from the Bangladesh Bureau of Statistics. Our sample gives a complete representation of the divisions we intended to cover. 

(Bangladesh Bureau of Statistics (BBS), 2022. Population & Housing Census 2022, Preliminary report. Ministry of Planning, Government of the People’s Republic of Bangladesh, 11. Available from: https://bbs.portal.gov.bd/sites/default/files/files/bbs.portal.gov.bd/page/b343a8b4_956b_45ca_872f_4cf9b2f1a6e0/2023-09-27-09-50-a3672cdf61961a45347ab8660a3109b6.pdf)

Comment: 4. Lines 85-86: without providing the reference, the authors wrote “while considering an 80% vaccine acceptance among parents based on the existing literature”

Response: Thank you for your observation. Referencing issues on lines 85-86 have been addressed in the revised manuscript. Please refer to line 97 in the revised manuscript to find the updated reference. 

Comment: 5. Lines 88-89: the authors wrote “of Bangladesh between June 28 and August 2, 2023, resulting in a total sample size of 2,151 after exclusions”. Your reader may be interested in knowing the overall response rate when invitation was issued to potential participants, total responses before the exclusion, and the exclusion criteria.

Response: We appreciate your feedback. The methodology section of the revised manuscript now includes details on the overall response rate upon invitation, total responses before exclusions, and the exclusion criteria. Please refer to lines 101-104 in the revised manuscript, which are now read as follows-

“The overall response rate, considering those who actively participated in the survey, was approximately 90%. Of the 2160 eligible participants who agreed to participate, 2151 completed the entire questionnaire (completion rate: 99.58%); incomplete questionnaires were excluded from the analysis.”

Comment: 6. Also, can the authors consider attaching an English version of their study questionnaire as supplement?

Response: Thank you for your feedback. The English version of the study questionnaire is attached in the supplemental material section (Please refer to S1 File).

Comment: 7. Lines 117-120, why is this statement (Second, a pre-test was conducted among 28 parents of 118 daughters aged between 9–15 years old prior to the start of the actual data collection. In order to make the questionnaire more understandable for the participants and to improve face- and construct validity, some questions were rewritten or clarified.) bolder than the rest of the manuscript?

Response: Thank you for bringing this to our attention. The statement in lines 117-120 has been revised for better clarity. In the revised manuscript, the lines are (Line 133-135) now read as follows-

 “Second, before commencing the actual data collection, a pre-test involving 28 parents of daughters aged between 9–15 years old was undertaken. To enhance participant comprehension and bolster both face- and construct validity, certain questions were revised or clarified in the questionnaire.”

Comment: 8. Pages 4, 5, and 6 have some redundancy of information. Please, try to scale down this. You may just reference the study questionnaire you attach to the supplement.

Response: Thank you for your suggestion. Although, based on your suggestion, we have included the questionnaire in the supplementary table, we believe the explanatory description provided on pages 4 to 6 is not redundant. If you note carefully, you may notice that we provided explanation on how some of the responses were grouped for analyses. Also, how we operationalized knowledge in the context of this research was explained. 

Comment: 9. How was knowledge scored?

Response: Thank you for the question. We explained it in the “knowledge and source of information about HPV and HPV vaccine” subsection of the method section. However, we noticed that the explanation was inadequate. Therefore, this section was modified to make it clear. The section now reads as follows-

“Knowledge was defined as the state of being aware of the term of interest in this study. Respondents' knowledge and source of information about HPV, HPV vaccine, cervical cancer, and cervical cancer-related vaccines were assessed through eight questions. First, they were asked if they had heard about HPV to investigate their awareness of HPV. If respondents answered affirmatively in the awareness-related question, their sources of information were sought. Knowledge about the HPV vaccine, cervical cancer, and cervical cancer-related vaccines was sought in a similar fashion. (lines 156 – 162).” 

Hence, the knowledge-related questions had mostly binary answers and didn’t require scoring.

Comment: 10. Also try to disaggregate the references for each questionnaire used to assess the various domains of the study, e.g. knowledge(reference), HBM (reference)

Response: Thank you for your suggestions. As explained earlier, knowledge meant ‘the state of being aware (or heard) about the term.’ If one was aware, we asked them about the sources of knowledge. Hence, this section of the questionnaire was built through brainstorming and discussion among the investigators. The literature review mainly helped us formulate the HBM-related question. We have now placed the references in the exact place to make it clear in the manuscript. Please check line 123. 

Comment: 11. On what basis did the authors classify monthly household income? One would expect the country’s minimum wage to be the reference for the income groups. Also, in no where in your manuscript did you spelt out your “BDT”

Response: “BDT” is the short form used globally to mean the monetary exchange note “Bangladeshi Taka”. Hence, we didn’t spell it out earlier. However, we have now spelt it in its first instance in the manuscript. Please check lines 207, 208.

Comment: 12. Line 136, The authors also mentioned “Academic institution type (government, private, madrasa)” as part of the study’s socio-demographic variables, which is important. However, they have not included this in the analysis without any explanation for dropping it.

Response: The academic institution type was the institute type of the daughters of the respondents. The question was not about the respondents (i.e., mother or father). We believe, the institute type of the respondents would have been valuable in the context of the vaccine acceptance by the respondent (not the adolescent girl). This is why this information was dropped. 

Comment: 13. In the regression model, why have the authors not considered exploring the eight divisions of the country as potential determinants?

Response: Thank you for your query. We examined the possibility of including division in the model. However, as you know, the underlying calculation of logistic regression involves combinations of categories of the different variables in the multivariable regression leading to the bifurcation of samples into incrementally smaller groups. Hence, when we add division, it produces unrealistic intervals in the multivariable regression. Therefore, we decided to exclude it from the final model and discuss divisional differences in follow-up papers. 

Comment: 14. There’s plethora of evidence supporting a parent’s/caregiver’s “level” of education with vaccination. However, I noticed the authors used “years of education” instead of “levels (primary, secondary, tertiary, university, e.t.c.”), which is more objective and informative. Why have the authors used their approach? I ask this because years of education does not necessarily mean level of education since one, for example, can decide to continue doing certificates after a primary or secondary education without progressing to a university degree.

Response: We choose years of education instead of level of education to incorporate the differences in institutional education among our participants. In our country, not everyone follows the same trajectory of education after secondary level. For example, some people directly go for a diploma in technical education rather than going for graduating from a university. Again, some choose the ‘Madrasa’ board, which again has two divisions (Aliya and Qaumi) instead of the national curriculum. Hence, we preferred to capture years of education instead of level of education. 

Comment: 15. Lines 287-290 in the discussion, the authors wrote: “When compared to prior research in similar contexts in India and China, where parental acceptance of the HPV vaccine was appro

---

## [Decision Letter · Decision Letter 1]

18 Jun 2024

PONE-D-23-37347R1Acceptance of Human Papillomavirus (HPV) vaccine among the parents of eligible daughters (9-15 years) in Bangladesh: a nationwide study using Health Belief ModelPLOS ONE

Dear Dr. Eva,

Thank you for submitting your manuscript to PLOS ONE. After careful consideration, we feel that it has merit but does not fully meet PLOS ONE’s publication criteria as it currently stands. Therefore, we invite you to submit a revised version of the manuscript that addresses the points raised during the review process.

We look forward to receiving your revised manuscript.

Kind regards,

Miquel Vall-llosera Camps

Senior Staff Editor

PLOS ONE

Journal Requirements:

Reviewers' comments:

Reviewer's Responses to Questions

**Comments to the Author**

1. If the authors have adequately addressed your comments raised in a previous round of review and you feel that this manuscript is now acceptable for publication, you may indicate that here to bypass the “Comments to the Author” section, enter your conflict of interest statement in the “Confidential to Editor” section, and submit your "Accept" recommendation.

Reviewer #2: All comments have been addressed

Reviewer #3: All comments have been addressed

Reviewer #5: (No Response)

2. Is the manuscript technically sound, and do the data support the conclusions?

Reviewer #2: Yes

Reviewer #3: Yes

Reviewer #5: Yes

3. Has the statistical analysis been performed appropriately and rigorously? 

Reviewer #2: Yes

Reviewer #3: Yes

Reviewer #5: Yes

4. Have the authors made all data underlying the findings in their manuscript fully available?

Reviewer #2: Yes

Reviewer #3: No

Reviewer #5: Yes

5. Is the manuscript presented in an intelligible fashion and written in standard English?

Reviewer #2: Yes

Reviewer #3: Yes

Reviewer #5: No

6. Review Comments to the Author

Reviewer #2: Authors have satisfactorily responded to my review comments. I recommend that this manuscript be accepted for publication.

Reviewer #3: 1. In Abstract Conclusion, the authors wrote “monthly income in the lowest quartile”, which is contrary to what they found from multivariable analysis. This should be middle-income since the income variable has four categories.

2. The Introduction is well-written and well-justified.

3. In the analysis section, how did the authors evaluate the fitness of their model? Consider to cite the references where applicable.

4. I think the description of the Results section is too wordy. Doesn’t it suffice to bring out some of the most salient point and refer the rest to the appropriate figure or table.

5. A significant number of studies on caregiver acceptance/hesitancy of childhood vaccines has reported being male (a father) as a significant determinant. In view of this, can your low number of male participants also be a limitation?

Best

Reviewer #5: It's good to see that the author has addressed quite a number of comments; however, there are a few lapses that need to be addressed:

1. Line 178 - "Perceived barrier" was repeated. Kindly address.

2. Line 191- You mentioned that R studio was used for data analysis. Please note that R studio is an integrated development environment (IDE) for the R programming language. It would be more accurate to mention that R Studio was used as an interface for data analysis, while R (the programming language) was used for statistical computations.

3. Throughout the data analysis section, there was no justification provided for the merging of responses for the HPV vaccine acceptance analysis. Therefore, please state the rationale behind merging the "No" and "Not sure" responses for the HPV vaccine acceptance analysis. This clarification assists readers in understanding the methodology and ensures transparency in the research process.

4. Lastly, consider including recommendations for future research. Providing suggestions for future research based on the study's findings, such as exploring interventions to address specific barriers to vaccine acceptance or conducting longitudinal studies to track acceptance trends, could be valuable.

7. PLOS authors have the option to publish the peer review history of their article (what does this mean?). If published, this will include your full peer review and any attached files.

Reviewer #2: No

Reviewer #3: **Yes: **Dr Sahabi Kabir Sulaiman

Reviewer #5: No

---

## [Author Response · Author response to Decision Letter 1]

8 Jul 2024

REBUTTAL LETTER

Reviewer #2: Authors have satisfactorily responded to my review comments. I recommend that this manuscript be accepted for publication.

Authors Response: Thank you for your thorough review of our manuscript.

Reviewer #3: 

1. In Abstract Conclusion, the authors wrote “monthly income in the lowest quartile”, which is contrary to what they found from multivariable analysis. This should be middle-income since the income variable has four categories.

Authors Response: Thank you for your insightful observation. We have revised the wording in the abstract conclusion for clarity. The updated text now reads:

Line 46-49: “Multiple factors such as younger age, urban residence, belonging to the middle income group, history of regular routine health check-ups, knowledge of cervical cancer, positive perception about benefits of the vaccine, and positive cues to actions were associated with HPV vaccine acceptance.”

2. The Introduction is well-written and well-justified.

Authors Response: Thank you for your comment.

3. In the analysis section, how did the authors evaluate the fitness of their model? Consider to cite the references where applicable.

Authors Response: The model performance was measured using Nagelkerke’s pseudo-R-squared (0.368), Receiver Operating Characteristics Curve (Area under the curve: 0.8471), and Hosmer-Lemeshow goodness of fit test (χ2 = 2145, df = 8, p <0.001). Although the model was not a good fit, we kept the model because of its overall significance over a null model (p<0.001) and because we were interested in identifying significant determinants of vaccine acceptance rather than the predictive accuracy of the overall model. 

We added this part in the methodology section. Please check lines 191 to 196. 

4. I think the description of the Results section is too wordy. Doesn’t it suffice to bring out some of the most salient point and refer the rest to the appropriate figure or table.

Authors Response: Thank you for your feedback regarding the Results section. We understand your concern about the level of detail. However, we believe that the current description provides necessary context and clarity for the reader. The detailed narrative is essential to fully convey the complexity of our findings and their implications. We have ensured that key points are highlighted and have referred to relevant figures and tables to support and illustrate the results. We hope this approach strikes an appropriate balance between detail and readability.

5. A significant number of studies on caregiver acceptance/hesitancy of childhood vaccines has reported being male (a father) as a significant determinant. In view of this, can your low number of male participants also be a limitation? Best

Authors Response: Thank you for the nice suggestion. We added this as a limitation in line numbers 404 – 405. 

Reviewer #5: It's good to see that the author has addressed quite a number of comments; however, there are a few lapses that need to be addressed:

1. Line 178 - "Perceived barrier" was repeated. Kindly address.

Authors Response: Thank you for your observation. We have addressed this comment in the revised manuscript. Please refer to lines 179-181 for the updated content.

2. Line 191- You mentioned that R studio was used for data analysis. Please note that R studio is an integrated development environment (IDE) for the R programming language. It would be more accurate to mention that R Studio was used as an interface for data analysis, while R (the programming language) was used for statistical computations.

Authors Response: Thank you for your comment. We have revised the wording for clarity and accuracy. The updated line (Line 199-201) now reads:

"R Studio (Version 2023.09.0+463) was used as an interface for data analysis, while R (the programming language) was used for statistical computations."

3. Throughout the data analysis section, there was no justification provided for the merging of responses for the HPV vaccine acceptance analysis. Therefore, please state the rationale behind merging the "No" and "Not sure" responses for the HPV vaccine acceptance analysis. This clarification assists readers in understanding the methodology and ensures transparency in the research process.

Authors Response: Thank you again for the detailed comment. We added clarification in the statistical analysis subsection of the methods section. Please check lines 189– 191.

4. Lastly, consider including recommendations for future research. Providing suggestions for future research based on the study's findings, such as exploring interventions to address specific barriers to vaccine acceptance or conducting longitudinal studies to track acceptance trends, could be valuable.

Authors Response: Thank you for your valuable suggestion. We have now incorporated a recommendations section for future research based on our study's findings. Please refer to lines 420-429 in the revised manuscript for these updates.

---

## [Decision Letter · Decision Letter 2]

5 Sep 2024

Acceptance of Human Papillomavirus (HPV) vaccine among the parents of eligible daughters (9-15 years) in Bangladesh: a nationwide study using Health Belief Model

PONE-D-23-37347R2

Dear Dr. Eva,

We’re pleased to inform you that your manuscript has been judged scientifically suitable for publication and will be formally accepted for publication once it meets all outstanding technical requirements.

Kind regards,

Rashidul Alam Mahumud, PhD, MCncrSc (Cancer Medicine), MPH, MSc,

Academic Editor

PLOS ONE

Additional Editor Comments (optional):

Reviewers' comments:

Reviewer's Responses to Questions

**Comments to the Author**

1. If the authors have adequately addressed your comments raised in a previous round of review and you feel that this manuscript is now acceptable for publication, you may indicate that here to bypass the “Comments to the Author” section, enter your conflict of interest statement in the “Confidential to Editor” section, and submit your "Accept" recommendation.

Reviewer #2: All comments have been addressed

Reviewer #3: All comments have been addressed

2. Is the manuscript technically sound, and do the data support the conclusions?

Reviewer #2: Yes

Reviewer #3: Yes

3. Has the statistical analysis been performed appropriately and rigorously? 

Reviewer #2: Yes

Reviewer #3: Yes

4. Have the authors made all data underlying the findings in their manuscript fully available?

Reviewer #2: Yes

Reviewer #3: No

5. Is the manuscript presented in an intelligible fashion and written in standard English?

Reviewer #2: Yes

Reviewer #3: Yes

6. Review Comments to the Author

Reviewer #2: (No Response)

Reviewer #3: My concerns have been addressed. The manuscripts has also significantly improved in this revision. I have no further comments.

7. PLOS authors have the option to publish the peer review history of their article (what does this mean?). If published, this will include your full peer review and any attached files.

Reviewer #2: No

Reviewer #3: **Yes: **Sahabi Kabir Sulaiman

---

## [Editor Report · Acceptance letter]

9 Sep 2024

PONE-D-23-37347R2 

PLOS ONE

Dear Dr. Eva, 

I'm pleased to inform you that your manuscript has been deemed suitable for publication in PLOS ONE. Congratulations! Your manuscript is now being handed over to our production team.

Kind regards, 

on behalf of

Dr. Rashidul Alam Mahumud 

Academic Editor

PLOS ONE